# Placental OLAH Levels Are Altered in Fetal Growth Restriction, Preeclampsia and Models of Placental Dysfunction

**DOI:** 10.3390/antiox11091677

**Published:** 2022-08-27

**Authors:** Natasha de Alwis, Sally Beard, Natalie K. Binder, Natasha Pritchard, Tu’uhevaha J. Kaitu’u-Lino, Susan P. Walker, Owen Stock, Katie Groom, Scott Petersen, Amanda Henry, Joanne M. Said, Sean Seeho, Stefan C. Kane, Stephen Tong, Lisa Hui, Natalie J. Hannan

**Affiliations:** 1Therapeutics Discovery and Vascular Function in Pregnancy Group, Mercy Hospital for Women, Heidelberg, VIC 3084, Australia; 2Mercy Perinatal, Mercy Hospital for Women, Heidelberg, VIC 3084, Australia; 3Northern Health, Epping, VIC 3076, Australia; 4Department of Obstetrics and Gynaecology, University of Melbourne, Parkville, VIC 3010, Australia; 5Translational Obstetrics Group, Mercy Hospital for Women, Heidelberg, VIC 3084, Australia; 6Liggins Institute, University of Auckland, Auckland 1023, New Zealand; 7Centre for Maternal Fetal Medicine, Mater Mothers’ Hospital, South Brisbane, QLD 4101, Australia; 8Discipline of Women’s Health, School of Clinical Medicine, UNSW Medicine and Health, University of New South Wales, Sydney, NSW 2052, Australia; 9Maternal Fetal Medicine, Joan Kirner Women’s & Children’s Sunshine Hospital, St Albans, VIC 3021, Australia; 10Women and Babies Research, Sydney Medical School-Northern, Faculty of Medicine and Health, University of Sydney, St Leonards, NSW 2065, Australia; 11Department of Maternal Fetal Medicine, Royal Women’s Hospital, Parkville, VIC 3052, Australia

**Keywords:** placenta, OLAH, fetal growth restriction, preeclampsia, fatty acid synthesis, pregnancy

## Abstract

Previously, we identified elevated transcripts for the gene Oleoyl-ACP Hydrolase (*OLAH*) in the maternal circulation of pregnancies complicated by preterm fetal growth restriction. As placental dysfunction is central to the pathogenesis of both fetal growth restriction and preeclampsia, we aimed to investigate OLAH levels and function in the human placenta. We assessed *OLAH* mRNA expression (qPCR) throughout pregnancy, finding placental expression increased as gestation progressed. *OLAH* mRNA and protein levels (Western blot) were elevated in placental tissue from cases of preterm preeclampsia, while OLAH protein levels in placenta from growth-restricted pregnancies were comparatively reduced in the preeclamptic cohort. *OLAH* expression was also elevated in placental explant tissue, but not isolated primary cytotrophoblast cultured under hypoxic conditions (as models of placental dysfunction). Further, we discovered that silencing cytotrophoblast *OLAH* reduced the expression of pro- and anti-apoptosis genes, *BAX* and *BCL2*, placental growth gene, *IGF2*, and oxidative stress gene, *NOX4*. Collectively, these findings suggest OLAH could play a role in placental dysfunction and may be a therapeutic target for mitigating diseases associated with this vital organ. Further research is required to establish the role of OLAH in the placenta, and whether these changes may be a maternal adaptation or consequence of disease.

## 1. Introduction

Fetal growth restriction and preeclampsia are serious disorders of pregnancy. Fetal growth restriction is characterized by an estimated fetal weight less than the tenth centile for gestational age, and is associated with an increased risk of stillbirth [1]. Preeclampsia is a complex condition characterized by new onset hypertension after 20 weeks’ gestation, accompanied by one or more of proteinuria, uteroplacental insufficiency, or maternal organ injury, and can lead to both maternal and perinatal morbidity or mortality [2]. Unfortunately, these conditions do not have treatments that can reverse disease pathogenesis. Hence, there is an urgent need for novel therapeutic strategies to treat these conditions.

Although preeclampsia and fetal growth restriction are two distinct disorders, they both feature a dysfunctional placenta. The placenta plays a crucial role in pregnancy, acting as the interface between the maternal and fetal systems, facilitating gas exchange, and the transfer of nutrients and waste. Abnormal spiral artery remodeling is a common feature of placental dysfunction, resulting in impaired placental blood supply [3], which can affect fetal growth. By improving our understanding of placental development and the specific mechanisms behind placental dysfunction, we can identify potential therapeutic targets for prevention and treatment.

Previously, we identified that transcripts for the gene Oleoyl-ACP Hydrolase (*OLAH*; also known as S-Acyl fatty acid synthase thioesterase or thioesterase II) were significantly elevated in the maternal circulation in pregnancies complicated by fetal growth restriction (with and without preeclampsia) [4]. Placenta-derived circulating RNAs are postulated to provide insight into the state of the placenta [5,6].

OLAH is an enzyme known for its role in fatty acid synthesis, where it acts alongside fatty acid synthase to preferentially produce shorter, medium chain fatty acids [7]. OLAH has been identified in mammary gland epithelial cells, particularly in the synthesis of the broad distribution of fatty acids in breast milk [7,8], and is elevated in bone marrow-derived mononuclear cells with rheumatoid arthritis [9]. Intriguingly, *OLAH* expression is described to be higher in human placental tissue compared to other species [10]. This suggests that OLAH may have a unique function in the human placenta and could potentially be involved in the pathogenesis of preeclampsia, a human-specific condition. Although fatty acids play an important role in the growth of the fetus, there is very little published on the role of OLAH in the placenta. Hence, the objective of this study was to examine OLAH levels in the placenta and to determine its role in placental development and disease.

## 2. Materials and Methods

### 2.1. FOX Study

As part of the FOX Study, maternal blood was collected from individuals whose pregnancies were complicated by preterm fetal growth restriction and controls with uncomplicated pregnancies, as previously described [4]. Samples were collected directly into PAXgene Blood RNA tubes (Pre-Analytix, Hombrechtikon, Switzerland) to preserve RNA quality, and processed according to the manufacturer’s instructions. All samples were collected prior to delivery, and after administration of corticosteroids.

Fetal growth restriction was defined as birthweight < 10th centile (www.gestation.net (accessed on 25 February 2022), Australian parameters) requiring iatrogenic delivery prior to 34 weeks’ gestation with evidence of uteroplacental insufficiency (asymmetrical growth + abnormal artery Doppler velocimetry ± oligohydramnios ± abnormal fetal vessel velocimetry). Fetal growth restriction due to congenital infection, chromosomal or congenital abnormalities, or multiple pregnancies were excluded. For the sub-analysis performed in this manuscript, cases of preterm fetal growth restriction were split into two groups; those where the mother was normotensive (normal blood pressure in pregnancy), and those where the mother had preeclampsia. Fetal hypoxic status was determined by measuring the pH of umbilical artery blood at birth, with fetal hypoxia defined as arterial pH < 7.2. All cases included in sub-analysis received antenatal corticosteroids. Patient characteristics are described in Appendix A.

### 2.2. Placental Tissue Collection

First trimester placental tissue was obtained from conceptus material collected at surgical terminations of singleton pregnancies (7–10 weeks’ gestation) under general anesthesia via curettage, or a combination of aspiration and curettage (according to the surgeon’s preference). Placental tissue was identified and isolated from conceptus material, then washed in phosphate-buffered saline (PBS). Placental tissue was transferred to RNAlater for 48 h, after which the tissue was snap frozen and stored at −80 °C for subsequent analysis. Patient characteristics are described in Appendix A.

Placentas were obtained from pregnancies complicated by early onset preeclampsia (requiring delivery ≤ 34 weeks’ gestation). Preeclampsia was defined according to the American College of Obstetricians and Gynecologists guidelines published in 2013 [11]. Placentas were obtained from cases of preterm fetal growth restriction (delivery ≤ 34 weeks’ gestation), defined as birth weight < 10th centile according to Australian population charts [12]. Cases associated with congenital infection, chromosomal or congenital abnormalities, multiple pregnancies, or preeclampsia were excluded.

Control healthy, term (delivery 37–40 weeks’ gestation) and preterm (delivery ≤ 34 weeks’ gestation) placentas were collected from normotensive pregnancies where a fetus of normal birth weight centile (>10th centile relative to gestation) was delivered. Placentas with evidence of chorioamnionitis, confirmed by placental histopathology, were excluded.

Term and preterm placental tissue were collected within 30 min of delivery. Preterm delivery of controls was predominantly for iatrogenic conditions including vasa previa, suspected placental abruption, and fetal anemia. Samples from four sites of the placenta were dissected, washed in ice cold PBS and preserved in RNAlater for 48 h, after which the tissue was snap frozen and stored at −80 °C for subsequent analysis. Patient characteristics are described in Appendix A.

Placentas were also obtained from healthy, normotensive term pregnancies (≥37 weeks’ gestation) at the elective cesarean section for explant dissection and cytotrophoblast isolation.

### 2.3. Collection and Culture of Placental Explants

Placental explants were collected with maternal and fetal surfaces removed by careful dissection. Three pieces of placenta were placed in each well of a 24-well plate (10–15 mg per well), and cultured in Gibco™ Dulbecco’s Modified Eagle Medium (DMEM; ThermoFisher Scientific, Scoresby, VIC, Australia) supplemented with 10% fetal calf serum (FCS; Sigma-Aldrich, St Louis, MO, USA) and 1% Antibiotic Antimycotic (AA; Life Technologies, Carlsbad, CA, USA). Explants were cultured under 8% O_2_, 5% CO_2_ at 37 °C overnight (16–18 h). After replacement with fresh media (DMEM/10% FCS/1% AA), explant tissue was cultured for a further 48 h under normoxic (8% O_2_, 5% CO_2_ at 37 °C) or hypoxic (1% O_2_, 5% CO_2_ at 37 °C) conditions. Following this, explant tissue was snap frozen and stored at −80 °C for subsequent analysis.

### 2.4. Primary Cytotrophoblast Isolation and Hypoxia Treatment

Primary human cytotrophoblast cells were isolated from healthy, term placentas at an elective cesarean section as previously described [13]. The cytotrophoblast cells were plated in media (DMEM/10% FCS/1% AA) on fibronectin (10 µg/mL; BD Bioscience, San Jose, CA, USA) coated culture plates. Viable cells were incubated under, 8% O_2_, 5% CO_2_ at 37 °C overnight to equilibrate and allow adhesion to cell culture plate. After replacement with fresh media (DMEM/10% FCS/1% AA), cytotrophoblasts were either cultured for a further 48 h under normoxic (8% O_2_, 5% CO_2_ at 37 °C) or hypoxic (1% O_2_, 5% CO_2_ at 37 °C) conditions or treated as described below. After 48 h, the cells were collected for RNA extraction.

### 2.5. Silencing OLAH in Primary Cytotrophoblast Cells

Short interfering (si)RNAs designed against *OLAH* (M-004796-01-0005; Dharmacon, Lafayette, CO, USA) or a pre-tested negative siRNA (Qiagen, Valencia, CA, USA) were combined with lipofectamine (RNAiMax; Invitrogen, Waltham, WA, USA) in Optimem media (ThermoFisher Scientific) at 10 nM, and allowed to complex for 20 min at room temperature. This siRNA complex was then added to overnight equilibrated cytotrophoblasts in fresh media (DMEM/10% FCS) in a dropwise manner. The cells were then incubated for a further 48 h under normoxic (8% O_2_, 5% CO_2_ at 37 °C) or hypoxic (1% O_2_, 5% CO_2_ at 37 °C) conditions. Following this, media and cell lysates were collected for subsequent analysis.

### 2.6. MTS Cell Viability Assay

Cell viability was assessed after siRNA treatment using an MTS assay. CellTiter 96-AQueous One Solution (Promega, Madison, WI, USA) was used according to the manufacturer’s instructions. Optical density was measured using a Bio-Rad X-Mark Microplate Spectrophotometer (Hercules, CA, USA) and Bio-Rad Microplate Manager 6 software.

### 2.7. Real Time Polymerase Chain Reaction (RT-PCR)

Total RNA was extracted from whole blood using the PAXgene^®^ Blood miRNA Kit (Pre-Analytix) as described previously [4]. RNA was extracted from placental tissue and cytotrophoblast cells using the Qiagen RNeasy Mini Kit following the manufacturer’s instructions. The RNA was quantified using a Nanodrop 2000 spectrophotometer (ThermoFisher Scientific, Waltham, MA, USA) or LVis Plate for FluoStar Omega Microplate Reader (BMG Labtech, Mornington, VIC, Australia). Extracted RNA was converted to cDNA using the Applied Biosystems^TM^ High-Capacity cDNA Reverse Transcription Kit, following the manufacturer’s instructions on the iCycler iQ5 (Bio-Rad) or MiniAmp Thermal Cycler (ThermoFisher Scientific). Quantitative Taqman PCR (with primers purchased from Life Technologies) was performed to quantify mRNA expression of *OLAH* (Hs00217864_m1), *PGF* (Hs00182176_m1), *HMOX1* (Hs01110250_m1), *NOX4* (Hs00418356_m1), *GCLC* (Hs00155249_m1), *NLRP3* (Hs00918082_m1), *SPINT1* (Hs00173678_m1), *BAX* (Hs00180269_m1), *BCL2* (Hs00608023_m1), *EGFR* (Hs01076078_m1), *IGF2* (Hs04188276_m1), *NQO1* (Hs00168547_m1) and *TXN* (Hs00828652_m1). The stability of reference genes was confirmed for each sample type and used appropriately; for blood *YWHAZ* (Hs01122454_m1), *B2M* (Hs00187842_m1) and *GUSB* (Hs00939627_m1), for cytotrophoblast *YWHAZ* (Hs01122454_m1) and for placental explants and placental tissue: *TOP1* (Hs00243257_m1) and *CYC1* (Hs00357717_m1). Taqman RT-PCR was performed on the CFX384 (Bio-Rad) with the following run conditions: 50 °C for 2 min; 95 °C for 10 min, 95 °C for 15 s, 60 °C for 1 min (40 cycles) or 95 °C for 20 s; 95 °C for 3 s, 60 °C for 30 s (40 cycles; Taqman Fast Advanced Master Mix).

The sFLT1 splice variants *sFLT1-i13* and *sFLT1-e15a* were measured in a SYBR PCR with SYBR Green Master mix (Applied Biosystems) using primers specific for each variant. The primers for *sFLT-i13* were 5′-ACAATCAGAGGTGAGCACTGCAA-3′ (forward) 5′-TCCGAGCCTGAAAGTTAGCAA-3′ (reverse), for *sFLT-e15a* 5-CTCCTGCGAAACCTCAGTG-3′ (forward) 5′-GACGATGGTGACGTTGATGT-3′ (reverse) and for *YWHAZ* (reference gene) 5′-GAGTCATACAAAGACAGCACGCTA-3′ (forward) 5′-TTCGTCTCCTTGGGTATCCGATGT-3′ (reverse). The SYBR PCR was run on the CFX384 (Bio-Rad), with 40 cycles of 95 °C for 21 s, then 60 °C for 20 min. All data were normalized to the appropriate reference gene as an internal control and calibrated against the average Ct of the control samples (2^−ΔΔCt^). All cDNA samples were run in duplicate.

### 2.8. Western Blot Analysis

Protein lysates were extracted from placental tissue collected from pregnancies complicated by preeclampsia, fetal growth restriction, and preterm controls (≤34 weeks), as well as placental explant tissue exposed to hypoxia using RIPA lysis buffer containing proteinase and phosphatase inhibitors (Sigma Aldrich). Protein concentration was determined with Pierce™ BCA Protein Assay Kit (ThermoFisher Scientific, Waltham, MA, USA). Protein lysates (20 µg) were separated on 12% gels before transfer to PVDF membranes (Millipore; Billerica, MA, USA). Membranes were blocked with 1% bovine serum albumin (BSA; Sigma-Aldrich) prior to overnight incubation with the primary anti-OLAH antibody (diluted 1:250 in 2% BSA/TBS-T; HPA037948, Sigma-Aldrich). Following this, membranes were incubated with a secondary anti-rabbit antibody (diluted 1:2500 in 2% BSA; W401, Promega, Madison WI, USA) for 1 h. Bands were visualized using a chemiluminescence detection system (GE Healthcare Life Sciences, Singapore) and ChemiDoc Imaging System (Bio-Rad). Β-actin acted as the loading control, (diluted 1:2500 in 5% skim milk; 51255, Cell Signalling Technology, Danvers, MA, USA). Relative densitometry was measured using Image Lab software 6.0.1 (Bio-Rad).

### 2.9. Enzyme Linked Immunosorbent Assay (ELISA)

Soluble fms-like tyrosine kinase-1 (sFLT1) and placental growth factor (PGF) secretion were measured in cytotrophoblast conditioned culture media using the DuoSet Human VEGF R1/FLT-1 kit and Human PlGF DuoSet ELISA kit (R&D systems by Bioscience, Waterloo, Australia), respectively, according to manufacturer’s instructions. Optical density was measured using a Bio-Rad X-Mark microplate spectrophotometer and Bio-Rad Microplate Manager 6 software.

### 2.10. Statistical Analysis

All in vitro experiments were performed with technical triplicates and repeated with a minimum of three different patient samples. Data were tested for normal distribution and for comparisons between two groups either an unpaired *t*-test or Mann–Whitney test was used as appropriate. For comparisons between three or more groups, a Kruskal–Wallis test was used with Dunn’s correction for multiple comparisons. All data are expressed as mean ± SEM. *p* values < 0.05 were considered significant. Statistical analysis was performed using GraphPad Prism software 8 (GraphPad Software, Inc.; San Diego, CA, USA).

## 3. Results

### 3.1. Circulating OLAH Transcripts Are Not Significantly Altered between Cases of Normotensive Fetal Growth Restriction, Compared to Cases of Preeclampsia with Growth Restriction

Using next-generation sequencing, we previously identified that *OLAH* transcripts are elevated in the maternal circulation in pregnancies complicated by fetal growth restriction, approximately half of which had preeclampsia [4]. Here, using data from the same sample set, we sought to evaluate whether those pregnancies that were also complicated by preeclampsia had further altered levels of circulating *OLAH* transcripts, compared to pregnancies complicated by fetal growth restriction alone (normotensive).

The samples that were used to measure *OLAH* in the preeclampsia cohort were from patients who had significantly higher BMI and delivered at an earlier gestational age, but were matched for other maternal and perinatal characteristics (Appendix A). We found that *OLAH* mRNA transcripts were not further significantly altered in the maternal circulation when pregnancies were also complicated by preeclampsia (plus growth restriction) compared to normotensive pregnancies with fetal growth restriction (Figure 1A). Further, we sub-analyzed these cases of fetal growth-restricted pregnancies to assess whether fetal acidemia had an impact on circulating *OLAH* transcripts, where umbilical artery blood pH < 7.2 indicated acidosis (pH ≥ 7.2 not acidotic). We identified that *OLAH* transcripts were not altered with fetal acidemia/hypoxia (Figure 1B).

### 3.2. OLAH Expression across Gestation

We next investigated whether *OLAH* was expressed in the human placenta. Placental samples were collected from the first trimester (7–10 weeks’ gestation), second trimester (24–28 weeks’ gestation), and third trimester (38–39 weeks’ gestation) (Appendix A). *OLAH* expression was detectable in all placental samples collected across gestation, but the expression was very low in three of the first trimester samples (close to the lower limit of detection). *OLAH* expression was significantly elevated in second trimester (*p* = 0.0032) and third trimester (*p* = 0.0025) placental samples compared to first trimester samples. There was no difference in expression between the second and third trimester placental samples (Figure 2).

### 3.3. OLAH Levels Are Significantly Altered in Placental Tissue from Cases of Preterm Preeclampsia and Fetal Growth Restriction

Noting that *OLAH* transcripts are highly increased in the circulation of cases of preeclampsia and growth restriction [4], we next examined OLAH levels in the pathological placenta, as placental dysfunction is central to the pathophysiology of both conditions. We assessed *OLAH* expression and protein in placental tissue from individuals with preterm preeclampsia and fetal growth restriction, compared to preterm control placental tissue (Appendix A).

Placental mRNA expression of *OLAH* was significantly increased in preterm preeclampsia (without fetal growth restriction) (*p* = 0.0004) compared to preterm control, but was not altered in fetal growth restriction alone, or preeclampsia with fetal growth restriction (Figure 3A). *OLAH* mRNA expression did not differ significantly between the pathological groups.

Placental OLAH protein production in cases of fetal growth restriction and preeclampsia ± growth restriction was not significantly different from the variable OLAH protein production in the preterm control tissue (Figure 3B,C; full immunoblots presented in Appendix A). However, OLAH protein in the placental tissue from cases of growth restriction was significantly lower than that of the tissue from cases of preeclampsia without growth restriction (*p* = 0.0042; Figure 3B,C; full immunoblots presented in Appendix A). OLAH protein levels were not different in placental tissue from cases of preeclampsia that featured fetal growth restriction compared to the other pathological groups (Figure 3B,C; full immunoblots presented in Appendix A).

### 3.4. OLAH Expression Is Upregulated in Placental Explant Tissue Cultured under Hypoxia

To uncover the mechanisms behind altered OLAH levels in the pathological placenta, we next examined *OLAH* expression in placental tissue explants and isolated cytotrophoblast cells (unique to the placenta) under hypoxia (low oxygen), as a model of placental dysfunction. *OLAH* expression was detectable in both placental tissue and cytotrophoblast cells under hypoxic (1% O_2_) and normoxic conditions (8% O_2_). *OLAH* expression was significantly increased in placental explant tissue under hypoxia, compared to normoxic conditions (*p* = 0.0011; Figure 4A). We followed this with western blotting to examine OLAH protein in placental explant tissue, but we could not detect OLAH at the protein concentration used (data not shown). *OLAH* mRNA expression was not altered in primary cytotrophoblasts cultured under hypoxia (Figure 4B).

### 3.5. Silencing OLAH in Primary Cytotrophoblast Cells Does Not Alter Cell Survival

To elucidate how dysregulated OLAH levels could affect placental function, studies were performed using short interfering (si)RNAs to silence *OLAH* expression in isolated primary cytotrophoblast cells. Expression was silenced by approximately 90% under both normoxic (*p* < 0.0001; Figure 5A) and hypoxic conditions (*p* < 0.0001; Figure 5B), which was determined sufficient knockdown for further experiments. Silencing OLAH did not significantly alter cell survival assessed via MTS assay, under either normoxic or hypoxic conditions (Appendix A).

### 3.6. Silencing Cytotrophoblast OLAH Does Not Alter sFLT1 Secretion

Levels of anti-angiogenic factor sFLT1 are elevated, and angiogenic factor PGF are reduced in the circulation of individuals whose pregnancies are complicated by preeclampsia [14]. Here, we assessed whether OLAH regulates the expression and release of these factors from isolated cytotrophoblast, which would implicate OLAH in the regulation of angiogenesis in preeclampsia. Under hypoxia (mimicking conditions of placental dysfunction akin to that in preeclampsia), silencing cytotrophoblast *OLAH* did not alter the expression of the two sFLT1 transcript isoforms, *sFLT-e15a* (Figure 6A) or *sFLT-i13* (Figure 6B), nor secretion of the functional sFLT1 protein (Figure 6C). Silencing *OLAH* under hypoxia significantly reduced *PGF* mRNA expression (*p* = 0.0002; Figure 6D), but did not alter PGF secretion (Figure 6E).

We also assessed the effects of silencing cytotrophoblast *OLAH* on sFLT1 and PGF levels in normoxic conditions to evaluate a potential role for OLAH in normal placental function. Here, we found that silencing *OLAH* did not alter *sFLT-e15a* expression (Appendix A), but significantly increased *sFLT-i13* expression (*p* = 0.0252; Appendix A), though sFLT1 secretion was not altered (Appendix A). Silencing *OLAH* under normoxic conditions did not alter cytotrophoblast *PGF* expression (Appendix A) or secretion (Appendix A).

### 3.7. Effect of Silencing Cytotrophoblast OLAH on Expression of Apoptosis, Growth, Inflammation, and Oxidative Stress Genes

To further uncover pathways associated with dysregulation of placental OLAH, we assessed how silencing cytotrophoblast *OLAH* affected the expression of genes associated with key cellular pathways including apoptosis, growth, oxidative stress, and inflammation. Under hypoxia, silencing *OLAH* significantly downregulated the expression of pro-apoptosis gene, *BAX* (*p* = 0.0142; Figure 7A) and anti-apoptosis gene, *BCL2* (*p* < 0.0001; Figure 7B). Silencing cytotrophoblast *OLAH* also significantly reduced the expression of *IGF2* (*p* = 0.0114; Figure 7C), but not *EGFR* or *SPINT1* (Figure 7D,E), genes which are also associated with fetal growth [15]. Antioxidant genes *HMOX1*, *NQO1*, *TXN,* and *GCLC* were not significantly altered with *OLAH* knockdown (Figure 7F–I), but the oxidative stress gene, *NOX4* (*p* = 0.0084; Figure 7D), was significantly reduced. Inflammasome gene, *NLRP3*, was not significantly altered with cytotrophoblast *OLAH* knockdown under hypoxia (Figure 7).

Under normoxic conditions, silencing *OLAH* significantly reduced expression of *BCL2* (*p* = 0.0188; Appendix A), but did not significantly alter the expression of *BAX*, *IGF2*, *EGFR*, *SPINT1*, *HMOX1*, *NQO1*, *TXN*, *GCLC*, *NOX4* or *NLRP3* (Appendix A).

## 4. Discussion

In previous studies, our group demonstrated that *OLAH* transcripts were highly upregulated in the maternal circulation of individuals whose pregnancies were complicated by fetal growth restriction and preeclampsia [4]. In this study, we identified that *OLAH* is expressed in the placenta throughout gestation, and importantly, its levels are altered in the placenta in cases of fetal growth restriction and preeclampsia (without fetal growth restriction), and under hypoxic conditions. Further, we found that silencing cytotrophoblast *OLAH* expression can alter the expression of genes associated with apoptosis, fetal growth, and oxidative stress.

In a sub-analysis of the *OLAH* transcripts measured in the maternal circulation of pregnancies complicated by fetal growth restriction, we found no further changes when we split the cases by hypertensive status or fetal hypoxia. Circulating RNAs could give us an insight into the state of the placenta [5,6,16], or act as signaling molecules, taken up by the vasculature to alter function in the maternal system. These findings suggest that dysregulation of OLAH is common to both fetal growth restriction and preeclampsia and may be the result of shared pathophysiological pathways. Further, fetal hypoxia specifically is unlikely to be related to circulating *OLAH* levels. If the elevated *OLAH* transcripts in these conditions were indeed taken up by the vasculature, exerting their canonical function in fatty acid synthesis could have implications for cardiovascular health. In fact, medium chain fatty acids, which are the products of OLAH-catalyzed reactions, are being considered for their therapeutic benefits on metabolic health in cardiac disease [17].

We next investigated OLAH in the placenta, as increased OLAH levels in the pathological placenta could contribute to the elevated levels of circulating *OLAH* transcripts, especially as both fetal growth restriction and preeclampsia can feature a dysfunctional placenta. We first measured *OLAH* expression over gestation, finding generally that its expression appeared at its lowest in the first trimester (with some samples displaying *OLAH* expression at the lower limit of detection), but was increased in the second and third trimesters. Placental *OLAH* gene expression has previously been demonstrated to be highest at term gestation, albeit in small sample sizes [18]. This increase in expression as gestation progresses could be associated with the increased circulating lipid levels in pregnancy, especially with rapid lipid lysis that occurs closer to term [19], as OLAH is required for the preferred synthesis of medium chain fatty acids.

Intriguingly, we found that placentas collected from cases of preeclampsia, but not fetal growth restriction, had significantly elevated *OLAH* mRNA expression compared to preterm controls. This suggests that OLAH may be regulated distinctly between these two conditions. It is for this reason that we analyzed the samples from cases of preeclampsia featuring growth restriction independently.

Although *OLAH* mRNA expression was elevated with preeclampsia, OLAH *protein* was not significantly different compared to control. However, we note that while OLAH protein levels in each of the samples from the preeclampsia and growth restriction groups were clustered together, the OLAH protein levels in the preterm controls were quite variable, and thus could have confounded our results. This is a limitation of the preterm control samples, though we have tried to minimize confounding factors within this group, these are not perfectly healthy pregnancies as they have resulted in premature delivery. Nevertheless, we did interestingly find that OLAH protein in growth-restricted placental tissue was significantly lower than that in the placental samples collected from cases of preeclampsia. This again supports the idea that OLAH may be regulated distinctly in these two conditions. Dysregulation of OLAH protein in these two diseases may contribute to differences previously identified in the fatty acid profile of maternal and umbilical cord plasma in pregnancies affected by fetal growth restriction alone versus preeclampsia with fetal growth restriction [20].

Reduced OLAH in the placenta may contribute to impaired fetal growth. Fatty acids are postulated to be an energy source utilized by the placenta [21]. Further, the smaller fatty acids are, the more readily they pass through the placenta to enter fetal capillaries [22]. The implication is that if OLAH levels are reduced, fatty acid synthase will preferentially produce larger chain fatty acids, which move less easily to the fetus, potentially impairing fetal development [23,24]. As we have detected low OLAH protein in placentas from cases of fetal growth restriction, we suggest that OLAH could play an important role in disease pathogenesis and could be an important target for improving fetal growth.

In contrast, OLAH levels were high in placenta collected from pregnancies complicated by preterm preeclampsia. Similarly, another study found that cytotrophoblast cells isolated from placentas experiencing preeclampsia had significantly elevated *OLAH* mRNA, compared to preterm labor controls [25]. These findings suggest that preeclampsia is associated with increased placental OLAH.

Preeclampsia is associated with dyslipidemia [26,27,28,29,30]. However, most studies focus on circulating and placental levels of long-chain fatty acids, rather than medium-chain fatty acids. Hence, we cannot make a statement on what increased placental OLAH could mean in preeclampsia without further studies, except to suggest that this elevated OLAH may mean increased lipid processing into medium-chain fatty acids, which may move more readily to the fetus for fetal growth. Thus, in preeclampsia, elevated placental OLAH could be beneficial.

In this study, we found that placental hypoxia significantly increased *OLAH* mRNA expression in placental explant tissue, similar to our findings in placental tissue from cases of preeclampsia, suggesting that impaired oxygen delivery may cause OLAH dysregulation in preeclampsia. Interestingly, isolated cytotrophoblast cells, which are unique to the placenta, did not have significantly altered *OLAH* expression under hypoxia. This suggests that the source of placental OLAH dysregulation may be altered as the trophoblast differentiates to the syncytium, or perhaps is another cell type, such as stromal, vascular, or immune cells that are also part of the placenta. Investigating *OLAH* expression in these different cell types could improve understanding of the role of placental OLAH in disease pathology. However, placental hypoxia is just one aspect of the pathology of these conditions. Altered cytotrophoblast *OLAH* expression has been demonstrated in cells isolated from preeclamptic placentas [25], suggesting OLAH may still have an important role in cytotrophoblast cells.

We therefore went on to investigate a potential role for OLAH in placental cytotrophoblasts in the pathogenesis of fetal growth restriction and preeclampsia. Silencing *OLAH* in cytotrophoblasts did not alter the expression or secretion of sFLT1, suggesting that it is not involved in the regulation of this anti-angiogenic factor. Under normoxic conditions, silencing cytotrophoblast *OLAH* increased the expression of sFLT1 isoform *sFLT-i13,* but not isoform *sFLT-e15a*. However, this did not affect the secretion of sFLT1 protein, likely because *sFLT-e15a* is the predominant isoform [31,32]. Under hypoxic conditions, silencing *OLAH* significantly increased the mRNA expression of angiogenic *PGF;* however, intriguingly did not alter the secretion of PGF protein. This suggests there is post-transcriptional or post-translational regulation of PGF release. Further investigation of intracellular PGF protein would be necessary to elucidate this. Under normoxic conditions, silencing *OLAH* did not alter the expression or secretion of PGF. These findings suggest that *OLAH* dysregulation in the placenta is unlikely to be associated with changes in these anti-angiogenic and angiogenic factors that are synonymous with preeclampsia, and hence would not be a suitable target for controlling levels of these factors in disease.

However, we did find that silencing *OLAH* altered expression of genes associated with apoptosis, fetal growth, and oxidative stress. Under hypoxia, silencing cytotrophoblast *OLAH* expression reduced the expression of both *BAX* and *BCL2*, anti- and pro-apoptotic genes, respectively. The reduction of both means that the balance of these factors may not be altered, and total apoptosis may not be affected [33]. We also detected a significant reduction in *IGF2*; knockdown of IGF2 is associated with growth restriction in mice [34]. As OLAH protein is low in fetal growth restriction, it is possible that its reduction in disease could decrease IGF2 as well. Silencing *OLAH* did not alter *EGFR* or *SPINT1* expression, which are also associated with fetal growth. Expression of anti-oxidant genes *HMOX1*, *NQO1*, *TXN*, and *GCLC*, and inflammasome gene *NLRP3* were also unaltered, suggesting that cytotrophoblast OLAH does not regulate anti-oxidant pathways or the NLRP3 inflammasome under hypoxia. However, *NOX4*, a marker of oxidative stress was significantly reduced—this could be seen as a beneficial effect. However, accompanied by the reduction in *IGF2* expression, it is unclear whether a reduction in OLAH could be protective, or enhance placental dysfunction. Notably, under normoxic conditions, many of these genes were not dysregulated. The only gene we found to be dysregulated was *BCL2*, the anti-apoptotic gene, which suggests the loss of cytotrophoblast OLAH under normoxic conditions could be detrimental.

Further studies will evaluate correlations between OLAH levels and placental pathologies in both preterm and term placentas. Assessing pathways altered with *OLAH* overexpression may give further insight into the implications of *OLAH* dysregulation in the placenta, especially because placental OLAH is elevated in preeclampsia and with placental hypoxia. Additionally, investigating OLAH protein secretion would help validate whether the low OLAH levels in growth-restricted placentas are truly due to a reduction in OLAH protein production, or increased secretion. An important extension of this work would be to measure specific production of medium chain fatty acids in response to changes in placental OLAH. Assessing the canonical pathways of OLAH and its interactions (e.g.,: with NFκB and PPAR-α pathways) in placental tissue and cytotrophoblast cells would be an important next avenue of investigation.

## 5. Conclusions

Using first trimester, preterm, and term placental tissues and cells from normal and pathological pregnancies, this study identified that OLAH expression is altered over gestation, and may have a role in placental dysfunction. Further investigation is required to establish the precise role of OLAH in the placenta and cytotrophoblast. However, we have identified that this gene could play an important role in the pathogenesis of preeclampsia and fetal growth restriction, whether it is due to its role in fatty acid synthesis is still to be determined.

## Figures and Tables

**Figure 1 antioxidants-11-01677-f001:**
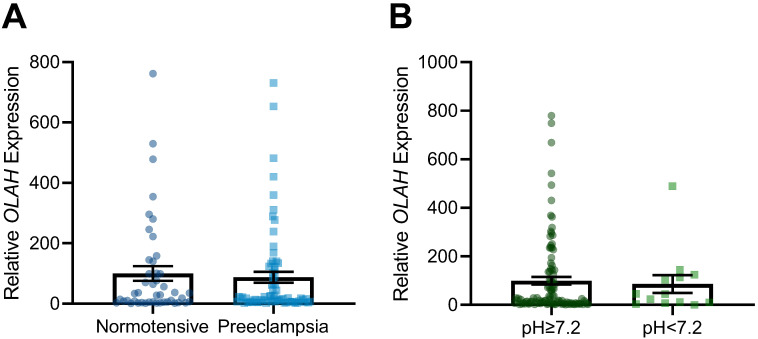
*OLAH* mRNA transcripts in the maternal circulation of pregnancies complicated by fetal growth restriction with or without preeclampsia or fetal acidemia. Transcript levels were measured via qPCR. *OLAH* transcripts were not significantly altered in the circulation of cases of preeclampsia with fetal growth restriction, compared to normotensive cases of fetal growth restriction (**A**). Cases of growth restriction where the fetus was delivered with umbilical cord artery pH < 7.2 did not have significantly altered *OLAH* transcripts in the maternal circulation compared to those with pH ≥ 7.2 (**B**). Data presented as relative expression, mean ± SEM. Normotensive *n* = 44, preeclampsia *n* = 71, case pH ≥ 7.2 *n* = 107, case pH < 7.2 *n* = 18.

**Figure 2 antioxidants-11-01677-f002:**
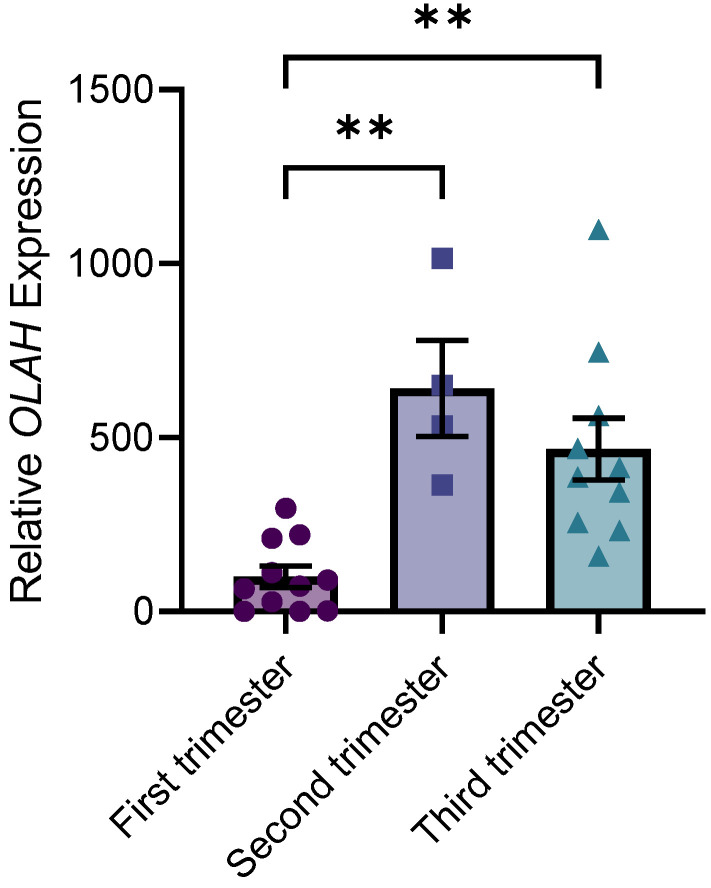
*OLAH* gene expression in placental samples collected in first, second, and third trimester. Gene expression was assessed via qPCR. *OLAH* expression was significantly increased in second and third trimester compared to first trimester. Expression in the second and third trimester was not significantly different. Data presented as relative expression, mean ± SEM. First trimester (7–11 weeks gestation) *n* = 11, second trimester (24–28 weeks gestation) *n* = 4, third trimester (38–39 weeks gestation) *n* = 10 placental samples. ** *p* < 0.01.

**Figure 3 antioxidants-11-01677-f003:**
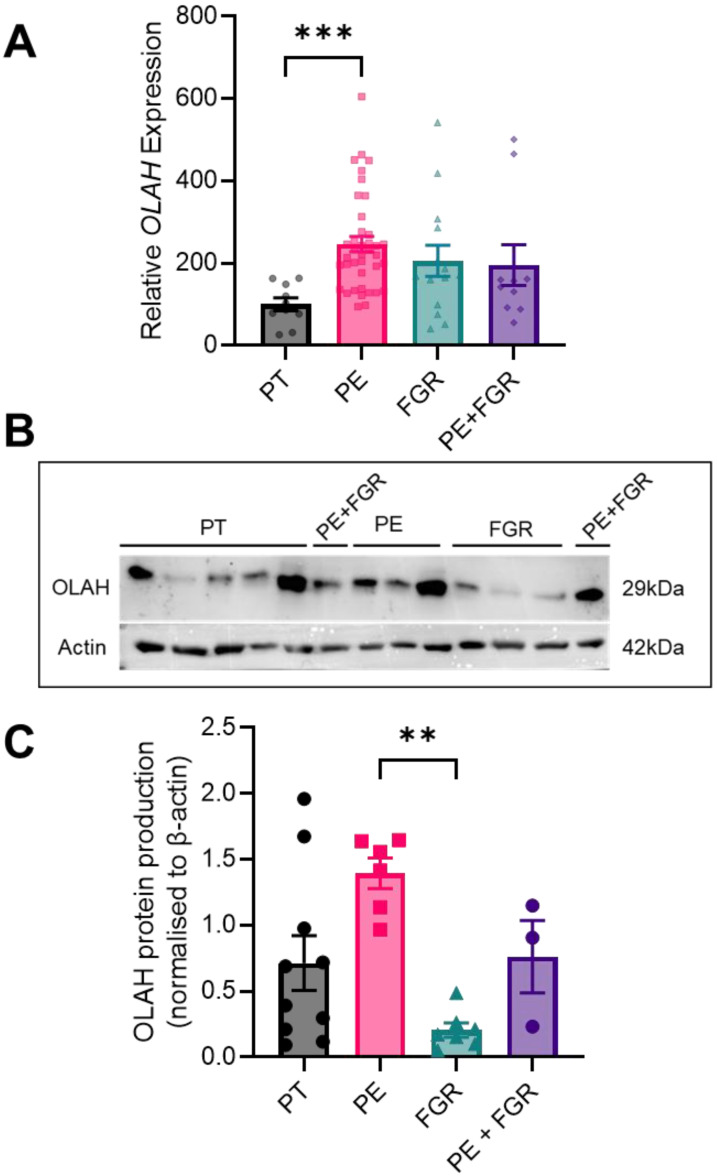
*OLAH* mRNA expression and protein production in placental tissue collected from cases of preterm preeclampsia and fetal growth restriction compared to preterm control. (**A**) Relative *OLAH* mRNA expression assessed via qPCR, (**B**) representative western blot image of OLAH protein production, and (**C**) densitometric analysis of western blot data. *OLAH* mRNA expression was significantly elevated in placental tissue collected from pregnancies complicated by preterm preeclampsia (PE) compared to preterm control (PT), but not altered in placenta from pregnancies complicated by fetal growth restriction (FGR), and both fetal growth restriction and preeclampsia (PE + FGR) (**A**). Densitometric analysis demonstrated that OLAH protein was not significantly altered in any group compared to preterm control tissue (**C**). However, OLAH protein was significantly reduced in placenta collected from pregnancies complicated by fetal growth restriction compared to samples collected from cases of preeclampsia (**C**). Data presented as mean ± SEM. Gene expression: PT *n* = 10, PE *n* = 39, FGR *n* = 14, PE + FGR *n* = 10. Protein production: PT *n* = 10, PE *n* = 6, FGR *n* = 7, PE + FGR *n* = 3. ** *p* < 0.01, *** *p* < 0.001. Full western blot images presented in Appendix A.

**Figure 4 antioxidants-11-01677-f004:**
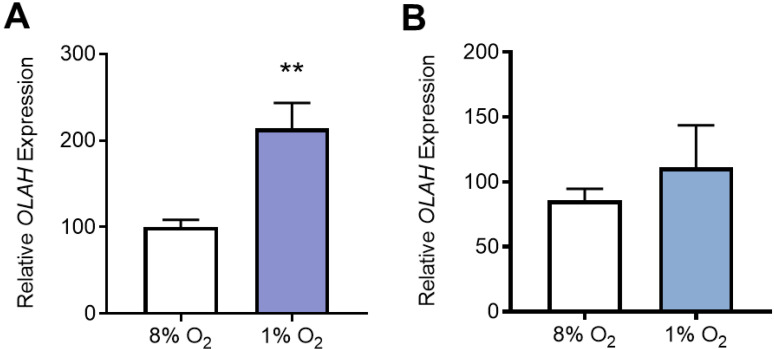
*OLAH* mRNA expression in term (**A**) placental explant tissue and (**B**) isolated cytotrophoblast cells under hypoxia. Gene expression was assessed by qPCR. *OLAH* expression was significantly higher in placental explant tissue under hypoxic conditions (1% O_2_) compared to tissue under normoxic conditions (8% O_2_) (**A**). There was no significant difference in *OLAH* transcripts in cytotrophoblast cells under hypoxic compared to normoxic conditions (**B**). Data presented relative to control; mean ± SEM. *n* = 4–5. ** *p* < 0.01.

**Figure 5 antioxidants-11-01677-f005:**
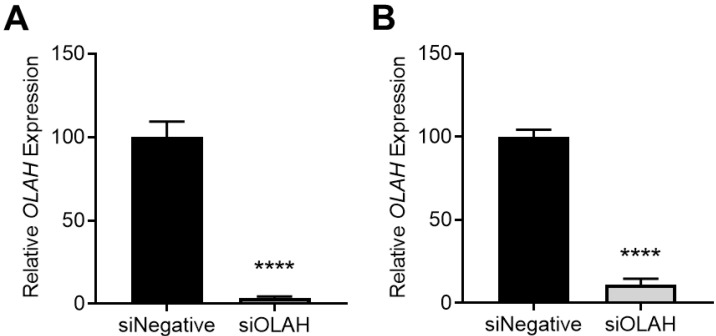
Knockdown of *OLAH* in cytotrophoblast cells under (**A**) normoxic and (**B**) hypoxic conditions. *OLAH* mRNA expression was significantly reduced with addition of siRNAs in both normoxic (8% O_2_) and hypoxic conditions (1% O_2_), assessed via qPCR. Data presented as relative mean ± SEM. *n* = 3. **** *p* < 0.0001.

**Figure 6 antioxidants-11-01677-f006:**
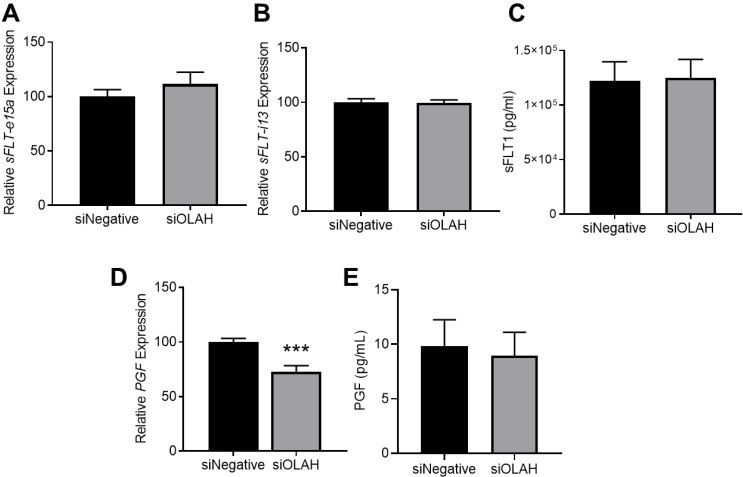
Effect of silencing *OLAH* on expression and secretion of anti-angiogenic sFLT1 and angiogenic PGF from cytotrophoblast cells under hypoxia (1% O_2_). Gene expression assessed via qPCR and protein secretion by ELISA. Silencing *OLAH* in cytotrophoblast cells did not alter expression of sFLT1 isoforms *sFLT-e15a* (**A**), *sFLT-i13* (**B**), nor sFLT1 secretion (**C**) under hypoxia. Silencing *OLAH* significantly reduced the expression of *PGF* (**D**), but did not alter PGF secretion (**E**) under hypoxia. Data presented as mean ± SEM. *n* = 3–4, each sample from a different patient. *** *p* < 0.001.

**Figure 7 antioxidants-11-01677-f007:**
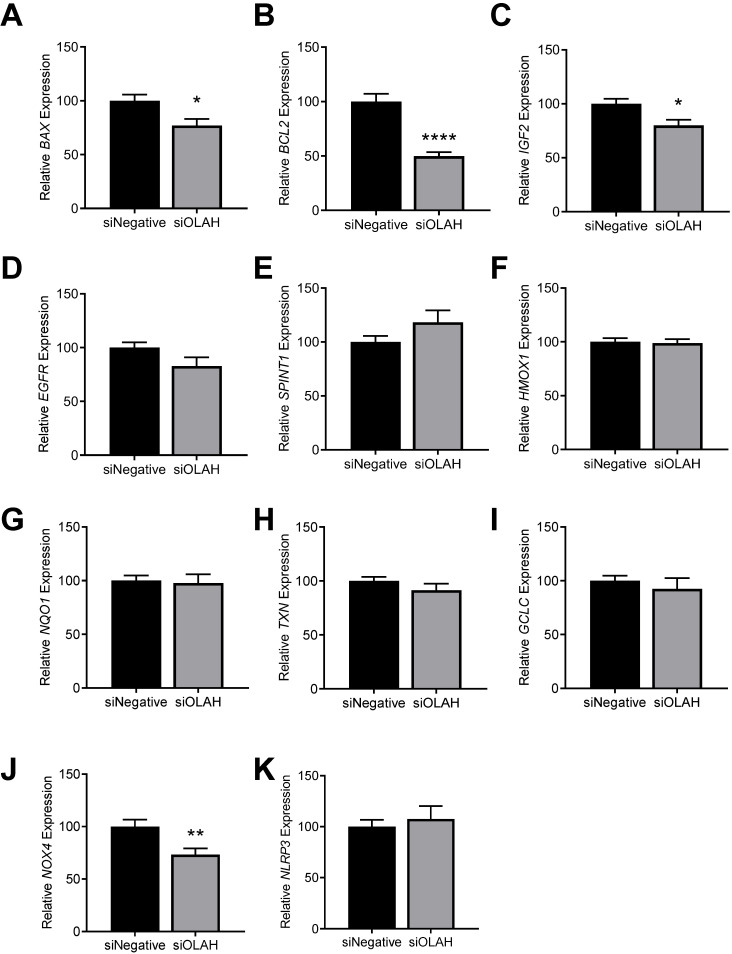
Effect of silencing cytotrophoblast *OLAH* on the expression of genes associated with apoptosis, growth, inflammation, and oxidative stress under hypoxia. Silencing *OLAH* significantly reduced expression of pro-apoptotic *BAX* (**A**) and anti-apoptotic *BCL2* (**B**) under hypoxia. Silencing *OLAH* significantly reduced cytotrophoblast expression of *IGF2* (**C**), but did not significantly alter *EGFR* (**D**), or *SPINT1* (**E**) expression, genes associated with fetal growth. Anti-oxidant genes *HMOX1, NQO1, TXN,* and *GCLC* (**F**–**I**, respectively) were not significantly altered with *OLAH* knockdown in cytotrophoblast cells under hypoxia. Oxidative stress gene, *NOX4,* was significantly reduced with *OLAH* knockdown under hypoxia (**J**), but inflammasome gene, *NLRP3* was not significantly altered (**K**). Results presented as relative change compared to control; mean ± SEM. *n* = 3, each sample from a different patient. * *p* < 0.05, ** *p* < 0.01, **** *p* < 0.0001.

## Data Availability

The data are contained within the article and Appendix A.

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
