# Peer review of "Placental OLAH Levels Are Altered in Fetal Growth Restriction, Preeclampsia and Models of Placental Dysfunction"

_antioxidants, 2022, doi:10.3390/antiox11091677_

Round 1

Reviewer 1 Report

The paper is fantastic!

It is well written and clear and shows scientific integrity. 

One question is: I note mothers with IUGR babies were given steroids, Could the steroids have altered maternal serum OLAH levels?

Also: another different line of thought to consider: could the fetal-placental unit in IUGR pregnancies actually be secreting some factor  to alter the maternal metabolic state in order to induce maternal release/ mobilization of shorter fatty acids to optimize fetal growth in the context of maternal malperfusion or a placenta that is under-grown/ structurally suboptimal. 

One method  I suggest for the further study is to take cases of preterm IUGR (and term IUGR)and controls and see if maternal OLA levels correlate in any way with placental pathology (especially maternal vascular malperfusion, accelerated maturation, distal villous hypoplasia). [ I personally think the real biological story is with IUGR and adding PE complicates the data and makes the biological story harder to follow]. 

Anyways: all this is aside the point. Your paper is almost flawless.

Reviewer 2 Report

Dear Authors,

In general, your study  is interesting and original that allows us to consider placental OLAH levels as a potential indicator of differences in the pathogenesis of preeclampsia and FGR and the possibility of further using it as a diagnostic marker of fatty acid metabolism in these pathologies. In addition, for your future research, it is possible to evaluate the interaction of OLAH with Nfkb and PPAR α in primary cytotrophoblast and placenta in PE and FGR.

I have a few questions in this regard.

Have you conducted a lipid metabolism survey in pregnant women? If yes, please indicate whether changes in triglycerides, cholesterol, HDL and LDL levels were observed?

Have you searched for correlations between OLAH expression, neonatal weight, and fetal Doppler data (in particular, PI MCA)?
